**Data Availability Statement:** All relevant data are within the paper and its Supporting Information files.

# COVID-19 vaccination intention among internally displaced persons in complex humanitarian emergency context, Northeast Nigeria

**Saheed Gidado**[1,2]*, **Melton Musa**[3], **Ahmed Ibrahim Ba'aba**[4], **Lilian Akudo Okeke**[5], **Patrick M. Nguku**[2], **Isa Ali Hassan**[6], **Ibrahim Muhammad Bande**[7], **Rabi Usman**[8], **Gideon Ugbenyo**[2], **Idris Suleman Hadejia**[9], **J. Pekka Nuorti**[1], **Salla Atkins**[1]

1 Health Sciences Unit, Faculty of Social Sciences, Tampere University, Tampere, Finland, 2 African Field Epidemiology Network, Nigeria Country Office, Abuja, Nigeria, 3 African Field Epidemiology Network, Borno State Field Office, Maiduguri, Nigeria, 4 African Field Epidemiology Network, Yobe State Field Office, Damaturu, Nigeria, 5 African Field Epidemiology Network, Adamawa State Field Office, Yola, Nigeria, 6 Borno State Ministry of Health, Maiduguri, Borno State, Nigeria, 7 Department of Disease Control and Immunization, Yobe State Primary Health Care Board, Damaturu, Yobe State, Nigeria, 8 Resolve to Save Lives, Abuja, Nigeria, 9 Department of Community Medicine, Ahmadu Bello University, Zaria, Kaduna State, Nigeria

* saheed.gidado@tuni.fi

## Abstract

Internally displaced persons (IDPs) are at high risk for COVID-19 transmission because of congested and unsanitary living conditions. COVID-19 vaccination is essential to build population immunity and prevent severe disease among this population. We determined the prevalence and factors associated with intention to accept COVID-19 vaccine among IDPs in Northeast Nigeria. This cross-sectional study, conducted during July–December 2022, included 1,537 unvaccinated IDPs from 18 IDPs camps. We performed a complex sample survey analysis and described participants' characteristics and vaccination intention with weighted descriptive statistics. We fitted weighted logistic regression models and computed adjusted odds ratios with 95% confidence intervals to identify factors associated with intention to accept COVID-19 vaccine. Of 1,537 IDPs, 55.4% were 18–39 years old, 82.6% were females, and 88.6% had no formal education. Among them, 63.5% (95% CI: 59.0–68.1) expressed intention to accept COVID-19 vaccine. Among the IDPs who intended to reject vaccine, 42.8% provided no reason, 35.3% had COVID-19 misconceptions, 9.5% reported vaccine safety concerns, and 7.4% felt no need. IDPs who perceived COVID-19 as severe (Adjusted Odds Ratio (AOR) = 2.31, [95% CI: 1.35–3.96]), perceived COVID-19 vaccine as effective (AOR = 4.28, [95% CI: 2.46–7.44]) and resided in official camps (AOR = 3.29, [95% CI: 1.94–5.56]) were more likely to accept COVID-19 vaccine. However, IDPs who resided 2 kilometers or farther from the nearest health facility (AOR = 0.34, [95% CI: 0.20–0.58]) were less likely to accept vaccine. Intention to accept COVID-19 vaccine among the IDPs was suboptimal. To improve vaccination acceptance among this population, health education and risk communication should be intensified to counter misinformation, strengthen vaccine confidence, and shape perception of COVID-19 severity, focusing on

**Funding:** Saheed Gidado, the first and corresponding author, received funding from Tampere University, Finland. The funding covered the cost of transportation, accommodation, and allowances during the fieldwork of the study. The funder had no role in study design, data collection and analysis, decision to publish, or preparation of the manuscript.

**Competing interests:** The authors have declared that no competing interests exist.

IDPs in unofficial camps. Appropriate interventions to deliver vaccines to remote households should be ramped up.

## Introduction

Coronavirus disease (COVID-19), an infectious disease caused by the novel severe acute respiratory syndrome coronavirus 2 (SARS-CoV-2), was first identified in December 2019 in Wuhan, China [1]. Following the rapid spread of the disease from its origin to many other countries worldwide, the World Health Organization (WHO), on 30 January 2020, declared the COVID-19 outbreak a public health emergency of international concern and characterized the outbreak as a pandemic on 11 March 2020 [2]. Since the pandemic started, over 774 million COVID-19 cases and more than 7 million deaths have been recorded globally [3]. In Africa, COVID-19 infections numbered over 9.5 million individuals, with over 175,000 deaths as of 25 February 2024 [3]. Apart from causing huge morbidities and mortalities, the pandemic has disrupted preventive, promotive, and curative healthcare services, particularly in Sub-Saharan Africa (SSA), causing indirect morbidities and mortalities [4, 5]. On 5 May 2023, WHO declared the end of COVID-19 global health emergency [6]. Nigeria recorded her first confirmed COVID-19 case on 27 February 2020. As of 3 March 2024, the country has recorded over 267,000 confirmed cases and more than 3,000 deaths nationwide [7]. About 46% of Nigeria's total population have been vaccinated with at least one dose of a COVID-19 vaccine, while 39% have been vaccinated with a complete primary series of a COVID-19 vaccine as of 26 November 2023 [8].

Generally, internally displaced persons (IDPs) are considered among high-risk populations for COVID-19 infection [9]. According to the United Nations Guiding Principles on Internal Displacement, IDPs are 'persons or groups of persons who have been forced or obliged to flee or to leave their homes or places of habitual residence, in particular as a result of or in order to avoid the effects of armed conflict, situations of generalized violence, violations of human rights or natural or human-made disasters, and who have not crossed an internationally recognized State border' [10]. In Northeast Nigeria, over two million persons are currently displaced due to a protracted complex humanitarian emergency in the region precipitated by the rebellious activities of armed insurgent groups [11]. Apart from the congested and unsanitary living conditions which predispose the IDPs to an elevated risk of COVID-19 transmission, the limited access of this population to health interventions, including COVID-19 testing and case management facilities, also increases their risk of mortality from COVID-19 infection [12, 13]. Sadly, IDPs face considerable challenges and difficulties in adhering to COVID-19 preventive measures owing to poor environmental and other contextual factors, further exacerbating the risk of disease transmission among this population [14, 15]. Although COVID-19 is no longer considered a public health emergency, there is a risk of the emergence of new variants that could be more transmissible and/or more severe with new surges in cases and deaths [16]. Against this backdrop, it is essential to optimize COVID-19 vaccination uptake among the IDPs to build population immunity against circulating virus variants and prevent severe COVID-19 infection, particularly given their elevated risk for COVID-19 infection, limited access to health facilities and interventions and the contextual obstacles to adopting COVID-19 preventive practices.

According to the WHO, COVID-19 vaccines are vital tools in the COVID-19 pandemic response [17]. However, the acceptance of COVID-19 vaccination among different

populations in various contexts has been negatively impacted by widespread controversy, mis-conceptions, and mistrust regarding vaccine safety and effectiveness [18, 19]. For instance, the global pooled COVID-19 vaccine acceptance was approximately 65% [20]. In Nigeria, previous research reported an estimated pooled COVID-19 vaccine acceptance of 20.0%– 58.2% among the adult population [21]. Furthermore, a study conducted among IDPs in Northeast Nigeria revealed that less than half (46.3%) of this population had received at least one dose of COVID-19 vaccine (Oxford/AstraZeneca or Moderna), while just about one-third (33.1%) had received two doses as of November 2022 [22].

In the wake of the extensive misinformation and mistrust that trailed the safety and effec-tiveness of COVID-19 vaccine, several studies were commissioned to investigate COVID-19 vaccination intention among different populations to enrich the literature and inform strate-gies to improve vaccine acceptance. However, the majority of these studies were conducted in stable, non-humanitarian contexts and focused on the general population, healthcare workers, students, and chronically ill patients, among others [23–30]. Evidently, there is a dearth of research on COVID-19 vaccination intention among IDPs in humanitarian emergency con-texts. As reported in the published literature, findings of research conducted among different population sub-groups in stable, non-humanitarian contexts cannot simply be extrapolated to inform disease prevention and control interventions in humanitarian emergency situations [31]. Accordingly, we conducted this study to determine the prevalence and factors associated with intention to accept COVID-19 vaccine among IDPs in Northeast Nigeria.

## Materials and methods

### Study setting

This study was conducted in the context of a complex humanitarian emergency in Borno, Adamawa, and Yobe States, otherwise known as the BAY States. These three states are among Nigeria's 36 States (and the Federal Capital Territory). Situated in Northeast Nigeria, the BAY States share international borders with Niger, Chad, and Cameroun. Borno and Adamawa States have a population of 6,111,462 and 4,902,055, respectively, while Yobe State has a popu-lation of 3,649,607 [32]. Altogether, these three states host an estimated 284 IDPs camps and camp-like structures with about 195,901 households and 855,020 IDPs [33]. Several IDPs camps are designated as official (formal) camps because of the presence of Government authorities and camp management structure. Besides the official camps, there are a number of unofficial (informal) camps and camp-like settings in these States. Health care services are pro-vided by health facilities located in the IDPs camps, particularly the official camps. However, several other facilities outside the camps, but mostly within the host communities, also serve the IDPs.

### Study design and data source

This cross-sectional study included a total of 1,537 IDPs aged 18 years and above, spread across 18 formal and informal IDPs camps in Borno, Adamawa, and Yobe States (BAY States) in Nigeria's northeast region. These IDPs were among the participants enrolled from July 25 to December 5, 2022, in a complex sample household survey to investigate COVID-19 knowl-edge, risk perception, and adherence to COVID-19 preventive measures among IDPs in the BAY States, Northeast Nigeria [22]. Due to practical and logistic considerations, participants enrolled in the complex sample household survey were sampled using a complex sampling methodology. As of their enrollment in the complex sample household survey, the IDPs included in the current study had not received any dose of COVID-19 vaccine based on his-tory (self-report). The socio-demographic and household data of these 1,537 IDPs, as well as

data regarding their COVID-19 knowledge, risk perception, and vaccination intention, were extracted from the database of the complex sample household survey. The sampling methodology and data collection approach for the complex sample household survey have been described elsewhere [22]. In brief, 18 IDP camps were selected across the BAY States, six in each state, based on key criteria derived from the International Organization for Migration (IOM) 's field assessment data, Northeast Nigeria Displacement Report Round 40 –March 2022 [33]. Each camp was further stratified into four distinct, well-delineated geographical strata. A simple random sampling technique was employed to proportionately sample households from each stratum based on the population size of the stratum. In each selected household, one household member aged 18 years or older was randomly selected and interviewed within the household to maintain privacy and confidentiality [22]. Data for the complex sample household survey were collected using a semi-structured data collection instrument uploaded on Open Data Kit (ODK). This data instrument is included as S1 File.

## Study variables

In the current study, the outcome variable was COVID-19 vaccination intention (intention to accept or reject COVID-19 vaccination). This variable was measured by asking the participants whether they would accept or reject COVID-19 vaccine. The binary response for the outcome variable was 'reject' (coded as 0) or 'accept' (coded as 1). For the purpose of data analysis, we designated 'reject' as the outcome reference category. The explanatory variables were grouped under different sections and included 1) socio-demographic data such as age, gender, highest education attained, marital status, religion, and occupation, 2) households and camps' characteristics data including estimated monthly household income, duration of residence in IDPs camps, household distance to the nearest health facility, camp status (official versus unofficial), and state of IDPs camp location, 3) participants' COVID-19-related knowledge, and 4) perception-related variables which included participants' perceived COVID-19 susceptibility, severity, and vaccine effectiveness. Table 1 presents an overview of the study variables and their categories.

## Assessment of COVID-19 knowledge

We assessed COVID-19-related knowledge using a 12-point assessment tool that included knowledge questions across three key domains, namely: 1) signs and symptoms of COVID-19, 2) mode of spread, and 3) COVID-19 preventive measures. This method is similar to the approach employed by other authors [34, 35]. One point was recorded for every correct response. Participants who scored 6 points and above (out of 12 points) were considered to have adequate knowledge, while those who scored less than 6 points possessed poor knowledge [36]. The assessment tool to evaluate COVID-19-related knowledge is included as S2 File.

## Assessment of perception variables

We assessed three variables related to participants' perception of COVID-19 and COVID-19 vaccine using an adapted Risk Behavior Diagnosis (RBD) scale [37]. These variables were 1) perceived susceptibility to COVID-19, 2) perceived severity of COVID-19, and 3) perceived effectiveness of COVID-19 vaccine. We used a 3-item scale to assess each of the three perception variables and evaluated each item on a 5-point Likert scale ranging from 1 (strongly disagree) to 5 (strongly agree). We utilized Cronbach's alpha coefficient of reliability to determine the internal consistency of the scales for the different perception scales. The Cronbach's alpha values for perceived susceptibility, perceived severity, and perceived vaccine effectiveness were 0.95, 0.93, and 0.96, respectively, indicating a very high internal consistency of these scales

**Table 1. Description and categories of study variables.**

| Variables | Description | Categories |
|---|---|---|
| **Outcome variable** | | |
| COVID-19 vaccination intention | Participants' intention to accept or reject COVID-19 vaccination | 0 = Reject (reference) |
| | | 1 = Accept |
| **Explanatory variables** | | |
| Age group (in years) | Participants' age group as of last birthday prior to the study | 0 = 18–29 (reference) |
| | | 1 = 30–39 |
| | | 2 = 40–49 |
| | | 3 = ≥ 50 |
| Sex | Participants' sex | 0 = Female (reference) |
| | | 1 = Male |
| Education level | Highest level of formal education attained | 0 = None (reference) |
| | | 1 = Primary |
| | | 2 = Post-primary* |
| Marital status | Current marital status of participants as of the time of the study | 0 = Never married (reference) |
| | | 1 = Presently married |
| | | 2 = Widowed |
| | | 3 = Others[†] |
| Religion | Participants' religious practices | 0 = Christianity (reference) |
| | | 1 = Islam |
| Monthly household income | Estimated monthly income of each household (in Nigerian Naira) | 0 = < 13,300 NGN[¶] (reference) |
| | | 1 = ≥ 13,300 NGN |
| Duration of residence in the camps | Length of stay or residence of participants in the IDPs camps | 0 = ≤ 5 years (reference) |
| | | 1 = > 5 years |
| Status of IDPs camp | Designation of IDPs camps as formal (official) or informal (unofficial)[Ψ] | 0 = Informal or unofficial (reference) |
| | | 1 = Formal or official |
| Households' distance to the nearest health facility | Distance (in kilometers) from the household to the nearest health facility | 0 = < 2km (reference) |
| | | 1 = ≥ 2km |
| COVID-19 -related knowledge | Assessment of participants' knowledge regarding COVID-19 | 0 = Poor (reference) |
| | | 1 = Satisfactory |
| Perceived susceptibility to COVID-19 | Overall assessment of participants' beliefs about their likelihood or risk of contracting COVID-19 | 0 = Low (reference) |
| | | 1 = High |
| Perceived severity of COVID-19 | Overall assessment of participants' beliefs about the significance or magnitude of COVID-19 threat, and the beliefs relating to the consequences of COVID-19 infection on their lives and livelihood | 0 = Low (reference) |
| | | 1 = High |
| Perceived effectiveness of COVID-19 vaccine | Overall assessment of participants' beliefs about the effectiveness and benefit of COVID-19 vaccine in protecting against serious illness or death from COVID-19 | 0 = Low (reference) |
| | | 1 = High |

* Secondary, tertiary

[†] Separated, divorced, cohabiting

[¶] Nigerian Naira (13,300 NGN = 30 US Dollars)

[Ψ] Formal (official) IDPs camps are recognized by Government authorities and are well-managed by designated camp management structures.

[38]. Further, we summed up the Likert scale scores for all three items in each perception scale to obtain a composite score ranging from 3 to 15 for each scale. We then performed a median split on the composite Likert scale scores of each perception scale to categorize participants

into low and high perception categories for the respective perception scales [39, 40]. The list of items for the different perception scales is included as S3 File.

## Data analysis and statistical methods

To account for the differential probabilities of selecting study participants using a complex sampling approach rather than a simple random sampling method, we performed a complex sample survey data analysis, consistent with the recommended data analysis approach for complex sampling surveys [41]. We employed an inverse probability weighting approach to determine the survey weight for each study participant. With this survey weight variable, we specified a survey design, created a survey design object, and performed a complex survey analysis to obtain weighted statistical estimates, proportions, standard errors, and confidence intervals [42]. We conducted a weighted univariate analysis to describe participants' socio-demographics, COVID-19 vaccination intention, and other characteristics. We used the Rao-Scott chi-square test, a design-adjusted version of the Pearson chi-square test, to determine the relationship between vaccination intention and key participants' characteristics in accordance with the standard statistical procedure for complex sample survey analysis [42]. Furthermore, we performed a weighted crude logistic regression analysis to determine the unadjusted association between each explanatory variable and vaccination intention, designating 'would reject vaccine' as the outcome reference category. Employing a backward stepwise regression approach, we fitted weighted multivariable logistic regression models to estimate the covariate-adjusted association between the explanatory variables and vaccination intention. Weighted adjusted odds ratios with 95% confidence intervals were computed. Additionally, we utilized the variance inflation factor (VIF) to examine multicollinearity among the explanatory variables in the multivariable models. The VIF values for all the variables were less than 5, indicating low multicollinearity among these variables [43]. Finally, the Akaike Information Criterion (AIC) was used to compare the quality of the different multivariable model candidates [44]. The model with the smallest AIC value was selected as the best-fit model for our data. Data were analyzed with R statistical and computing software version R-4.2.2. [45].

## Ethical approval and consent to participate

Approval for this study was granted by the National Health Research Ethics Committee of Nigeria (NHREC) [NHREC Approval Number: NHREC/01/01/2007-14/06/2022; NHREC Assigned Number: NHREC/01/01/2007]. In addition, we obtained the consent, approval, and permission of the Borno State Ministry of Health (SHREC Approval No: 45/2022), Adamawa State Ministry of Health (Approval No: ADHREC 05/07/2022/051), and Yobe State Ministry of Health and Human Services (Approval No: YB/MOH/HREC/04/22/010). Prior to the conduct of the original complex sample household survey, we explained the purpose, procedure, and benefit of the research in the local language to study participants and responded to their questions and concerns. Furthermore, we assured the participants of voluntary participation and the opportunity to withdraw from the study at any time without prejudice in line with the Helsinki Declaration [46]. Moreover, we assured and maintained confidentiality during and after the study. Owing to heightened insecurity in the research area, participants' fear of being persecuted, and low education level, we obtained verbal informed consent from the participants in the presence of legally authorized representatives. We documented their affirmation and consent to participate in the study in the researchers' notes and the data collection tool. Mindful of the vulnerability of our study participants, we were guided by key considerations of health research ethics in humanitarian contexts as described in the published literature [47].

### Inclusivity in global research

Additional information regarding the ethical, cultural, and scientific considerations specific to inclusivity in global research is included in the S1 Checklist.

## Results

### Socio-demographic characteristics

A total of 1,537 participants were enrolled. There was no missing data element in the extracted data file of these participants. The weighted median age of the participants was 36 years (95% CI: 35–40 years) with an interquartile range of 15 years. Table 2 shows the socio-demographic characteristics of the participants. Of the IDPs, 23.6% (95% CI: 19.7–28.0) were 18 to 29 years old, while 31.8% (95% CI: 27.4–36.0) were aged 30–39 years. Most of the IDPs were females: 82.6% (95% CI: 79.1–86.0), had no formal education: 88.6% (95% CI: 85.7–91.0), unemployed: 52.2% (95% CI: 47.4–57.0) and resided in households that earn less than 30 US dollars per month: 85.5% (95% CI: 82.3–89.0).

### Knowledge and perception regarding COVID-19

Among the IDPs, 25.7% (95% CI: 21.6–29.8) were considered to have adequate COVID-19 knowledge, 48.7% (95% CI: 44.0–53.4) perceived themselves to be susceptible to COVID-19, 43.2% (95% CI: 38.5–47.9) perceived COVID-19 as severe, while 56.7% (95% CI: 52.0–61.4) perceived COVID-19 vaccine as effective.

### COVID-19 vaccination intention

Of the 1,537 participants, 1,105 expressed intention to accept COVID-19 vaccine, corresponding to a weighted proportion of 63.5% (95% CI: 59.0–68.1), whereas 432 participants, with a weighted proportion of 36.5% (95% CI: 31.9–41.0) reported intention to reject COVID-19 vaccine. Table 3 presents the absolute numbers and weighted proportions of participants' COVID-19 vaccination intention by key characteristics. Intention to accept COVID-19 vaccine differed significantly by IDPs' marital status, household distance to the nearest health facility, and COVID-19-related knowledge. Additionally, intention to accept vaccine among the IDPs varied significantly by perceived susceptibility to COVID-19, perceived severity of COVID-19, and perceived effectiveness of COVID-19 vaccine.

### Reasons for intention to reject COVID-19 vaccine

Of the 432 participants who reported intention to reject COVID-19 vaccine, 110, corresponding to a weighted proportion of 42.8% (95% CI: 34.9–50.6), did not provide any reason for their intention to reject the vaccine. However, 35.3% (95% CI: 27.7–42.9) gave reasons related to myths, misconceptions, and misinformation, 9.5% (95% CI: 5.0–13.9) expressed fear and concerns about vaccine safety and side effects, while 7.4% (95% CI: 3.3–11.4) felt no need for COVID-19 vaccination (Table 4).

### Factors associated with intention to accept COVID-19 vaccine

Table 5 shows the results of the weighted crude and multivariable logistic regression analysis of factors associated with intention to accept COVID-19 vaccine. At crude analysis, marital status, COVID-19-related knowledge, perceived susceptibility to COVID-19, perceived severity of COVID-19, and perceived effectiveness of COVID-19 vaccine demonstrated significant positive association with intention to accept COVID-19 vaccine. In contrast, household

**Table 2. Socio-demographic characteristics of study participants.**

| Characteristics (N = 1537) | Number of participants | Weighted % (95% CI) |
|---|---|---|
| **Age group (years)** | | |
| 18–29 | 464 | 23.6 (19.7–28.0) |
| 30–39 | 464 | 31.8 (27.4–36.0) |
| 40–49 | 310 | 24.4 (20.3–28.0) |
| ≥ 50 | 299 | 20.2 (16.4–24.0) |
| **Sex** | | |
| Female | 1030 | 82.6 (79.1–86.0) |
| Male | 507 | 17.4 (14.0–21.0) |
| **Highest level of formal education attained** | | |
| None | 1134 | 88.6 (85.7–91.0) |
| Primary | 261 | 8.4 (5.9–11.0) |
| Post-primary (secondary, tertiary) | 142 | 3.0 (1.5–4.0) |
| **Marital Status** | | |
| Never married | 130 | 6.3 (4.1–9.0) |
| Presently married | 1206 | 70.1 (65.7–74.0) |
| Widowed | 117 | 11.0 (8.0–14.0) |
| Others* | 84 | 12.6 (9.4–16.0) |
| **Religion** | | |
| Christianity | 65 | 0.6 (0.1–1.0) |
| Islam | 1472 | 99.4 (98.9–100.0) |
| **Occupation** | | |
| Unemployed | 661 | 52.2 (47.4–57.0) |
| Farmers | 549 | 19.9 (16.2–24.0) |
| Traders/Business | 206 | 23.6 (19.5–28.0) |
| Artisan (skilled labourer) | 64 | 2.3 (0.9–4.0) |
| Students | 33 | 0.9 (0.08–2.0) |
| Others$^\Psi$ | 24 | 1.1 (0.1–2.0) |
| **Estimated monthly household income** | | |
| < 13,300 NGN$^\P$ | 1019 | 85.5 (82.3–89.0) |
| ≥ 13,300 NGN | 518 | 14.5 (11.3–18.0) |
| **Duration of residence in IDPs camp** | | |
| ≤ 5 years | 445 | 34.6 (30.1–39.0) |
| > 5 years | 1092 | 65.4 (60.9–70.0) |

* Separated, divorced, cohabiting

$^\Psi$ Civil servants, unskilled laborers, drivers

$^\P$ Nigerian Naira (13,300 NGN = 30 US Dollars)

distance to the nearest health facility was negatively associated with intention to accept vaccine. Adjusting for co-variates at multivariable logistic regression analysis, IDPs who perceived COVID-19 as a severe illness compared to those who did not (Adjusted Odds Ratio (AOR) = 2.31, [95% CI: 1.35–3.96]), and IDPs who perceived COVID-19 vaccine as effective compared to those who did not (AOR = 4.28, [95% CI: 2.46–7.44]), irrespective of whether they resided in formal or informal camps, were significantly more likely to accept COVID-19 vaccine. Furthermore, IDPs who resided in formal (official) camps compared to those who resided in informal (unofficial) camps (AOR = 3.29, [95% CI: 1.94–5.56]) were significantly more likely to accept COVID-19 vaccine. However, IDPs who resided 2 kilometers or farther from the nearest health facility (AOR = 0.34, [95% CI: 0.20–0.58]) were less likely to accept vaccine.

**Table 3. COVID-19 vaccination intention by key participants' characteristics.**

| Participants' characteristics (N = 1537) | COVID-19 vaccination intention | | | | Chi-squared[Ψ], p-value |
|---|---|---|---|---|---|
| | Would reject vaccine (N = 432) | | Would accept vaccine (N = 1105) | | |
| | Number of participants | Weighted % (95% CI) | Number of participants | Weighted % (95% CI) | |
| **Age group (years)** | | | | | |
| 18–29 | 131 | 24.6 (17.8–31.3) | 333 | 23.1 (18.2–28.0) | 0.50, 0.68 |
| 30–39 | 109 | 28.1 (21.0–35.3) | 355 | 33.9 (28.3–39.4) | |
| 40–49 | 91 | 25.9 (18.9–32.8) | 219 | 23.5 (18.5–28.5) | |
| ≥ 50 | 101 | 21.4 (14.9–27.9) | 198 | 19.5 (14.8–24.2) | |
| **Sex** | | | | | |
| Female | 300 | 80.1 (73.9–86.4) | 730 | 83.9 (79.8–88.1) | 1.04, 0.31 |
| Male | 132 | 19.9 (13.6–26.1) | 375 | 16.1 (11.9–20.2) | |
| **Highest educational level attained** | | | | | |
| None | 340 | 89.3 (84.6–94.0) | 794 | 88.2 (84.6–91.7) | 0.19, 0.83 |
| Primary | 69 | 8.3 (4.1–12.6) | 192 | 8.5 (5.4–11.6) | |
| Post-primary (secondary, tertiary) | 23 | 2.4 (0.1–4.6) | 119 | 3.3 (1.5–5.2) | |
| **Marital status** | | | | | |
| Never married | 43 | 10.6 (5.8–15.5) | 87 | 3.8 (1.7–6.0) | 5.94, <0.001 |
| Presently married | 313 | 66.3 (58.8–73.8) | 893 | 72.3 (66.9–77.6) | |
| Widowed | 42 | 6.0 (2.3–9.6) | 75 | 14.0 (9.8–18.1) | |
| Others* | 34 | 17.1 (11.1–23.1) | 50 | 9.9 (6.3–13.6) | |
| **Religion** | | | | | |
| Christianity | 15 | 0.9 (0.4–2.2) | 50 | 0.5 (0.3–0.6) | 0.77, 0.38 |
| Islam | 417 | 99.1 (97.7–99.8) | 1055 | 99.5 (99.4–99.7) | |
| **Estimated monthly household income** | | | | | |
| < 13,300 NGN[¶] | 303 | 89.6 (85.0–94.2) | 716 | 83.2 (78.9–87.4) | 3.61, 0.06 |
| ≥ 13,300 NGN | 129 | 10.4 (5.8–15.0) | 389 | 16.8 (12.6–21.1) | |
| **Duration of residence in the camp** | | | | | |
| ≤ 5 years | 110 | 30.9 (23.5–38.2) | 335 | 36.7 (31.0–42.4) | 1.49, 0.22 |
| > 5 years | 322 | 69.1 (61.8–76.5) | 770 | 63.3 (57.6–69.0) | |
| **Household distance to the nearest health facility** | | | | | |
| < 2km | 315 | 57.8 (50.0–65.7) | 802 | 84.5 (80.4–88.7) | 38.66, <0.001 |
| ≥ 2km | 117 | 42.2 (34.3–50.0) | 303 | 15.5 (11.3–19.6) | |
| **Status of IDPs camp** | | | | | |

(*Continued*)

**Table 3.** (Continued)

| Participants' characteristics (N = 1537) | | COVID-19 vaccination intention | | | | Chi-squared$^{\Psi}$, *p*-value |
|---|---|---|---|---|---|---|
| | | Would reject vaccine (N = 432) | | Would accept vaccine (N = 1105) | | |
| | | Number of participants | Weighted % (95% CI) | Number of participants | Weighted % (95% CI) | |
| | Informal | 291 | 46.7 (38.9–54.6) | 816 | 38.1 (32.5–43.7) | 3.12, 0.08 |
| | Formal | 141 | 53.3 (45.4–61.1) | 289 | 61.9 (56.3–67.5) | |
| **COVID-19-related knowledge** | | | | | | |
| | Poor | 341 | 80.1 (73.8–86.4) | 734 | 71.0 (65.7–76.3) | 4.29, 0.04 |
| | Satisfactory | 91 | 19.9 (13.6–26.2) | 371 | 29.0 (23.7–34.3) | |
| **Perceived susceptibility to COVID-19** | | | | | | |
| | Low | 311 | 72.3 (65.2–79.4) | 324 | 39.3 (33.5–45.0) | 43.03, <0.001 |
| | High | 121 | 27.7 (20.6–34.8) | 781 | 60.7 (54.9–66.5) | |
| **Perceived severity of COVID-19** | | | | | | |
| | Low | 308 | 74.3 (67.5–81.2) | 519 | 46.8 (40.9–52.7) | 30.64, <0.001 |
| | High | 124 | 25.7 (18.8–32.5) | 586 | 53.2 (47.3–59.1) | |
| **Perceived effectiveness of COVID-19 vaccine** | | | | | | |
| | Low | 320 | 70.4 (63.2–77.7) | 306 | 27.7 (22.4–33.0) | 73.39, <0.001 |
| | High | 112 | 29.6 (22.3–36.8) | 799 | 72.3 (67.0–77.6) | |

$^{\Psi}$ Rao-Scott chi-square test–a design-adjusted version of Pearson chi-square test

* Separated, divorced, cohabiting

⁵ Nigerian Naira (13,300 NGN = 30 US Dollars)

## Discussion

In this cross-sectional study, we investigated COVID-19 vaccination intention among unvaccinated IDPs in Northeast Nigeria and identified the factors associated with intention to accept COVID-19 vaccine as a protective measure against severe disease. The study provides epidemiological evidence to improve COVID-19 vaccination uptake, particularly among crisis-

**Table 4. Reasons for intention to reject COVID-19 vaccine.**

| Reasons for intention to reject COVID-19 vaccine (N = 432) | Number of participants | Weighted % (95% CI) |
|---|---|---|
| Related to myths, misconceptions, and misinformation | 142 | 35.3 (27.7–42.9) |
| Related to fear and concern about vaccine safety and side effects | 103 | 9.5 (5.0–13.9) |
| No felt need | 49 | 7.4 (3.3–11.4) |
| Related to lack of consent from the husband or household head | 24 | 3.7 (0.8–6.6) |
| No reason given | 110 | 42.8 (34.9–50.6) |
| Others* | 4 | 1.4 (0.5–3.3) |

*Poor knowledge, inadequate information

**Table 5. Results of weighted logistic regression analysis of factors associated with intention to accept COVID-19 vaccine.**

| Participants' characteristics | Univariable model$^\Psi$ | | Multivariable model$^\Psi$ | |
|---|---|---|---|---|
| | COR[†] | 95% CI | AOR[‡] | 95% CI |
| **Age group (years)** | | | | |
| 18–29 (reference) | 1.00 | | 1.00 | |
| 30–39 | 1.28 | 0.75–2.19 | 1.11 | 0.55–2.23 |
| 40–49 | 0.97 | 0.55–1.69 | 0.82 | 0.40–1.68 |
| ≥ 50 | 0.97 | 0.54–1.75 | 0.58 | 0.25–1.35 |
| **Sex** | | | | |
| Female (reference) | 1.00 | | 1.00 | |
| Male | 1.29 | 0.79–2.13 | 1.94 | 0.98–3.84 |
| **Highest educational level attained** | | | | |
| None (reference) | 1.00 | | 1.00 | |
| Primary | 1.03 | 0.52–2.05 | 0.89 | 0.39–1.99 |
| Post-primary (secondary, tertiary) | 1.44 | 0.46–4.53 | 1.14 | 0.24–5.36 |
| **Religion** | | | | |
| Christianity (reference) | 1.00 | | 1.00 | |
| Islam | 1.93 | 0.43–8.66 | 0.39 | 0.11–1.39 |
| **Estimated monthly household income** | | | | |
| <13,000 NGN[¶] (reference) | 1.00 | | 1.00 | |
| ≥ 13,000 NGN | 1.74 | 0.98–3.10 | 1.23 | 0.58–2.63 |
| **Duration of residence in IDPs camp** | | | | |
| ≤ 5 years (reference) | 1.00 | | 1.00 | |
| > 5 years | 0.77 | 0.50–1.17 | 0.95 | 0.53–1.70 |
| **Marital status** | | | | |
| Never married (reference) | 1.00 | | 1.00 | |
| Presently married | 3.01* | 1.38–6.58 | 7.46* | 3.08–18.03 |
| Widowed | 6.48* | 2.32–18.14 | 10.39* | 3.69–29.24 |
| Others[$] | 1.61 | 0.64–4.06 | 3.54 | 1.22–10.22 |
| **COVID-19-related knowledge** | | | | |
| Poor (reference) | 1.00 | | 1.00 | |
| Satisfactory | 1.64* | 1.02–2.64 | 1.14 | 0.62–2.10 |
| **Perceived susceptibility to COVID-19** | | | | |
| Low (reference) | 1.00 | | 1.00 | |
| High | 4.04* | 2.63–6.20 | 1.72 | 1.00–2.98 |
| **Perceived severity of COVID-19** | | | | |
| Low (reference) | 1.00 | | 1.00 | |
| High | 3.30* | 2.14–5.08 | 2.31* | 1.35–3.96 |
| **Perceived effectiveness of COVID-19 vaccine** | | | | |
| Low (reference) | 1.00 | | 1.00 | |
| High | 6.22* | 4.02–9.63 | 4.28* | 2.46–7.44 |
| **Status of IDPs camp** | | | | |
| Informal (reference) | 1.00 | | 1.00 | |
| Formal | 1.43 | 0.96–2.12 | 3.29* | 1.94–5.56 |
| **Household distance to nearest health facility** | | | | |
| < 2km (reference) | 1.00 | | 1.00 | |

(*Continued*)

**Table 5.** (Continued)

| Participants' characteristics | | Univariable model[Ψ] | | Multivariable model[Ψ] | |
|---|---|---|---|---|---|
| | | COR[†] | 95% CI | AOR[‡] | 95% CI |
| | ≥ 2km | 0.25* | 0.16–0.39 | 0.34* | 0.20–0.58 |

[Ψ] Intention to reject vaccine designated as the reference category

[†] Weighted Crude Odds Ratio

[‡] Weighted Adjusted Odds Ratio

[¶] Nigerian Naira (13,300 NGN = 30 US Dollars)

[§] Separated, divorced, cohabiting

* Statistically significant at $P < 0.05$

affected, vulnerable populations in complex humanitarian emergency contexts. Study participants were mostly females, unemployed, had no formal education, and predominantly resided in households that earned less than 30 US dollars income per month. Nearly two-thirds of the IDPs expressed intention to accept COVID-19 vaccine. Whereas perception of COVID-19 severity, perception of COVID-19 vaccine effectiveness, and status of IDPs camps were positively associated with intention to accept COVID-19 vaccine, household distance to the nearest health facility was negatively associated with intention to accept vaccine.

About 64% of IDPs in this study expressed intention to accept COVID-19 vaccine. This proportion is higher than the COVID-19 vaccination acceptance rate of 56.7% reported among migrants, refugees, and foreign workers in a systematic review and meta-analysis [48], but lower than 89.6% obtained among Syrian refugees in Jordan [49], 70.3% reported among refugees in the United States [50], and 70% reported among migrant and refugee groups in a systematic review and meta-analysis [51]. Compared to other populations, our result is similar to the figure obtained among the general population in China (63.3%), but lower than those reported among Bangladeshi adults (74.6%), Pakistani university students (72.5%), and the general population in Northwest Nigeria (72.4%) [23, 24, 52, 53]. Although the COVID-19 vaccination population proportion required to achieve herd immunity is still a subject of scientific debate, the proportion of IDPs in our study with the intention to accept vaccine fell short of a minimum of 90% vaccination proportion reported by some global health experts [54, 55]. Additionally, prior research have shown that the proportion of individuals with the intention to accept vaccine is usually lower than the eventual vaccination coverage [56, 57]. Presumably, the foregoing narrative indicates a gap in COVID-19 vaccination acceptance among the IDPs and underscores the need for targeted interventions to improve vaccination acceptance and uptake among this high-risk population.

Among the IDPs who would reject COVID-19 vaccine in this study, more than half gave reasons related to COVID-19 myths and misconceptions, concerns about vaccine safety and side effects, and no felt need as the justifications for their intention to reject vaccine, similar to findings reported by other authors [27, 58, 59]. Our results reflect the pervasive misinformation and erroneous beliefs regarding COVID-19 and COVID-19 vaccine at national and global scenes. For instance, COVID-19 was regarded as a hoax in many quarters across the world [60], while several individuals, particularly in sub-Saharan Africa, believe that COVID-19 was a ploy to reduce the world's population [61]. Moreover, the safety and effectiveness of COVID-19 vaccine were subject of intense controversy among several population subgroups worldwide [62, 63]. Importantly, about 43% of IDPs who intend to reject vaccine provided no reason for their standpoint. We posit that understanding the reasons for vaccine rejection will guide evidence-based interventions to address its root cause and improve vaccination

acceptance among this population. In this regard, we suggest a qualitative inquiry to gain a deeper insight into the reasons for intention to reject COVID-19 vaccine in this context.

Our study indicated that IDPs who perceived COVID-19 as severe were significantly more likely to accept COVID-19 vaccine compared to those who did not, similar to the findings of other studies [64, 65]. This result aligns with the underlying constructs of the Health Belief Model, the Protection Motivation Theory, and several other models of health behavior [66, 67]. Fundamentally, when individuals perceive a disease (or other health conditions) as severe, they eagerly strive to overcome various forms of physical and socio-cultural impediments so as to access preventive and therapeutic care to maintain health and well-being. Given that less than half (43.2%) of the IDPs in the current study perceived COVID-19 as severe, our result provides the basis for health authorities to ramp up context-specific, culturally appropriate COVID-19 educational and risk communication activities tailored to the needs of the IDPs to shape disease risk perception and promote informed decision-making on COVID-19 vaccination among this population to increase vaccination acceptance and uptake.

We demonstrated that IDPs who perceived COVID-19 vaccine as effective compared to those who did not were significantly more likely to accept COVID-19 vaccine. This result is consistent with the findings of previous research conducted in Ethiopia, Ghana, and Nigeria [26, 28, 68]. Existing literature indicates that perceived vaccine effectiveness (response efficacy) motivates individuals' adoption of COVID-19 protective health behavior, including vaccination acceptance and uptake to avert COVID-19 threats [69]. Despite the widespread controversy and misinformation regarding the safety and effectiveness of COVID-19 vaccine among the general public [70], several authors have reported that perceived COVID-19 vaccine effectiveness is a major antidote to vaccine hesitancy [71, 72]. Furthermore, previous research have shown that providing information on COVID-19 vaccine effectiveness improves vaccination acceptance and uptake [73]. Seemingly, this evidence favors the feasibility of enhancing COVID-19 vaccination uptake by crafting and disseminating tailored information, education, and communication (IEC) messages to particularly emphasize the effectiveness of the COVID-19 vaccine, as documented in the published literature [74, 75].

Furthermore, IDPs who resided in formal (official) camps were significantly more likely to accept COVID-19 vaccine compared to those who resided in informal (unofficial) camps. Typically, formal camps in our study setting are officially recognized by the Government agencies responsible for emergency management and humanitarian affairs. As such, IDPs who reside in formal camps apparently benefit from healthcare, social, and other humanitarian support from Governmental and Non-Governmental Organizations (NGOs). For instance, with the support of Governmental agencies and NGOs, many health facilities in formal camps offer basic health care services, antenatal care, immunization, and other health interventions at a free or heavily subsidized cost. Moreover, facilities in formal camps served as the programmatic hubs for the planning and implementation of COVID-19 pandemic response interventions, including IEC activities, COVID-19 vaccination, case detection and reporting, among others. We believe that the facilitated access of IDPs in formal camps to COVID-19 vaccination, as well as their exposure to COVID-19 IEC at the health facilities, could partly explain their higher likelihood to accept COVID-19 vaccine compared to IDPs residing in informal camps.

Our study revealed an inverse association between household distance to health facility and intention to accept vaccine. We presume that this finding is partly related to the perceived financial challenges confronting the IDPs. For instance, about 85% of households in this study earn less than 30 US dollars per month. With this paltry income, it is possible that a large number of IDPs might not be able to afford the cost of transportation and logistics from remote areas to health facilities. We opine that this factor could negatively influence vaccination

intention among this population and probably explain the observed inverse association between household distance to health facilities and intention to accept vaccine. Our result resonates with the findings of previous research, which identified distance as a major barrier to the uptake of vaccination and other healthcare services at health facilities among the general population [76–79]. Furthermore, IDPs who reside farther from health facilities could have limited access to facility-driven IEC activities about COVID-19 and COVID-19 vaccine. Therefore, it is important for local health authorities to prioritize IDPs residing farther from health facilities for COVID-19 IEC activities to create awareness and improve vaccination acceptance. Moreover, appropriate interventions, such as a mobile vaccination strategy to reach and deliver vaccines to IDPs in remote households, should be implemented and/or ramped up.

This study draws strength from at least two key methodological considerations. Firstly, the IDPs studied were recruited via a stratified sampling approach. Based on existing literature, this sampling approach probably increased the precision of the statistical estimates obtained in our study [80]. Secondly, we performed complex survey data analysis, incorporating survey weights to obtain weighted statistical estimates and confidence intervals. We believe that this weighted data analysis approach likely enhances the validity and generalizability of the study findings. However, we recognize certain limitations of this study. Data for the study were mostly retrospective and self-reported. As such, we admit that the potential for recall bias and misinformation cannot be ruled out. Additionally, we recognize a limitation regarding the scope of the study; including a qualitative component in the study would have expanded its scope and provided additional insights, particularly into the reasons for vaccine rejection.

## Conclusions

As the global health community consolidates the gains of COVID-19 pandemic control, our study provides evidence to improve COVID-19 vaccination acceptance among IDPs in Northeast Nigeria. About two-thirds of these IDPs expressed intention to accept COVID-19 vaccine. Misconceptions, vaccine safety concerns and no felt need were the prominent reasons for vaccine rejection. Perceived COVID-19 severity, perceived vaccine effectiveness, and residing in formal camps were positively associated with intention to accept COVID-19 vaccine. However, household distance from the nearest health facility demonstrated a negative association with intention to accept vaccine. The study underscores the need for local health authorities and other relevant stakeholders to ramp up context-specific, culturally appropriate COVID-19 health education, risk communication, and public enlightenment activities tailored to the needs of the IDPs to counter misinformation, correct misconceptions, shape disease risk perception, strengthen vaccine confidence and promote informed-decision making on COVID-19 vaccination, particularly among IDPs in unofficial camps. Authorities should implement or ramp up appropriate interventions, including mobile vaccination strategy to deliver vaccines to IDPs residing in remote households. Further qualitative research should deep dive into the reasons for vaccination rejection to permit a better understanding of this phenomenon.

## Supporting information

**S1 Checklist. Inclusivity in global research.**
(DOCX)

**S1 File. Data instrument.**
(PDF)

**S2 File. Assessment algorithm for COVID-19 related knowledge.**
(PDF)

**S3 File. List of items used to assess perceived COVID-19 susceptibility, severity, and vaccine effectiveness.**
(PDF)

**S1 Data. Research data underlying the findings reported in the manuscript.**
(XLSX)

## Acknowledgments

The authors acknowledge the effort and support of the Ministries of Health of Borno, Adamawa, and Yobe States. We appreciate the leadership of our study IDPs camps and health partner organizations in these states. Additionally, we sincerely acknowledge Adam Attahiru, Toman Emmanuel, Alhaji Dalatu, Musa Ashafa, Hassan Umar Kamfut, Fatima Mohammed El-Yakub, Ahmed Gambo Ibbi, and Bulama Maina Yaro for their technical support.

## Author Contributions

**Conceptualization:** Saheed Gidado, J. Pekka Nuorti, Salla Atkins.

**Data curation:** Gideon Ugbenyo.

**Formal analysis:** Saheed Gidado, Gideon Ugbenyo.

**Funding acquisition:** J. Pekka Nuorti.

**Investigation:** Melton Musa, Ahmed Ibrahim Ba'aba, Lilian Akudo Okeke, Isa Ali Hassan, Ibrahim Muhammad Bande, Rabi Usman, Idris Suleman Hadejia, J. Pekka Nuorti.

**Methodology:** Melton Musa, Ahmed Ibrahim Ba'aba, Lilian Akudo Okeke, Patrick M. Nguku, Isa Ali Hassan, Ibrahim Muhammad Bande, Rabi Usman.

**Project administration:** Melton Musa, Patrick M. Nguku, Salla Atkins.

**Resources:** Patrick M. Nguku, Idris Suleman Hadejia, Salla Atkins.

**Software:** Gideon Ugbenyo.

**Supervision:** Saheed Gidado, Melton Musa, Ahmed Ibrahim Ba'aba, Lilian Akudo Okeke, Patrick M. Nguku, Isa Ali Hassan, Ibrahim Muhammad Bande, Rabi Usman, Idris Suleman Hadejia.

**Validation:** Melton Musa, Ahmed Ibrahim Ba'aba, Lilian Akudo Okeke, Patrick M. Nguku, Isa Ali Hassan, Ibrahim Muhammad Bande, Rabi Usman, Idris Suleman Hadejia.

**Writing – original draft:** Saheed Gidado, J. Pekka Nuorti, Salla Atkins.

**Writing – review & editing:** Saheed Gidado, Melton Musa, Ahmed Ibrahim Ba'aba, Lilian Akudo Okeke, Patrick M. Nguku, Isa Ali Hassan, Ibrahim Muhammad Bande, Rabi Usman, Gideon Ugbenyo, Idris Suleman Hadejia, J. Pekka Nuorti, Salla Atkins.

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
