## [Decision Letter · Decision Letter 0]

26 Feb 2024

PONE-D-24-02466COVID-19 vaccination intention among internally displaced persons in complex humanitarian emergency context, Northeast NigeriaPLOS ONE

Dear Dr. Gidado,

Thank you for submitting your manuscript to PLOS ONE. After careful consideration, we feel that it has merit but does not fully meet PLOS ONE’s publication criteria as it currently stands. Therefore, we invite you to submit a revised version of the manuscript that addresses the points raised during the review process.

The manuscript is well-written and timely, particularly given the invaluable insights garnered from navigating the challenges posed by the COVID-19 pandemic. It is imperative to document such experiences, especially concerning vulnerable populations, to inform future healthcare strategies effectively.

However, to provide a comprehensive understanding of the paper's context, it is essential to delineate the impact of the COVID-19 pandemic in Africa and Nigeria, particularly concerning morbidity, mortality, vaccination rates, and other pertinent information. This comparative analysis would elucidate the specific challenges faced by internally displaced persons (IDPs) and set the backdrop for the study's relevance.

While the paper employs various statistical methodologies, it lacks clarity regarding the percentage of IDPs, regardless of their formal or informal status, willing to accept the COVID-19 vaccine. Additionally, the issue of proximity to health facilities was highlighted, warranting further exploration, particularly in comparison to the general population. Also as COVID 19 is no longer a pandemic, can this also be reflected in the paper.

Further elaboration on the statistical methodologies employed would enhance the paper's accessibility, especially for readers with limited statistical proficiency. Clarification and additional information on these methodologies would mitigate confusion and facilitate a deeper understanding of the findings.

Lastly, it is advisable to limit references to publicly available "published work" and consider the feedback provided by Dr. Edina and Bassey. Their insights could further refine and strengthen the manuscript.

We look forward to receiving your revised manuscript.

Kind regards,

Sylvester Maleghemi Maleghemi, MD,MSC, DTH&M, MBA

Academic Editor

PLOS ONE

Journal Requirements:

"Saheed Gidado, the first and corresponding author, received funding from Tampere University, Finland. The funding covered the cost of transportation, accommodation, and allowances during the fieldwork of the study. The funder had no role in study design, data collection and analysis, decision to publish, or preparation of the manuscript."

We note that one or more of the authors is affiliated with the funding organization, indicating the funder may have had some role in the design, data collection, analysis or preparation of your manuscript for publication; in other words, the funder played an indirect role through the participation of the co-authors. If the funding organization did not play a role in the study design, data collection and analysis, decision to publish, or preparation of the manuscript and only provided financial support in the form of authors' salaries and/or research materials, please do the following:

a. Review your statements relating to the author contributions, and ensure you have specifically and accurately indicated the role(s) that these authors had in your study. These amendments should be made in the online form.

b. Confirm in your cover letter that you agree with the following statement, and we will change the online submission form on your behalf: 

“The funder provided support in the form of salaries for authors [insert relevant initials], but did not have any additional role in the study design, data collection and analysis, decision to publish, or preparation of the manuscript. The specific roles of these authors are articulated in the ‘author contributions’ section.

4. In the online submission form, you indicated that [Data cannot be shared publicly because the study was conducted in the context of armed conflict induced humanitarian emergency triggered by the insurgent activities of Non-State Armed Groups (NSAG) against the Nigerian Government. This setting is associated with enormous security concerns and risk, particularly for the internally displaced persons - the study population. However, the data are available from the corresponding author on request.]. 

Reviewers' comments:

Reviewer's Responses to Questions

**Comments to the Author**

1. Is the manuscript technically sound, and do the data support the conclusions?

Reviewer #1: Yes

Reviewer #2: Partly

2. Has the statistical analysis been performed appropriately and rigorously? 

Reviewer #1: Yes

Reviewer #2: Yes

3. Have the authors made all data underlying the findings in their manuscript fully available?

Reviewer #1: No

Reviewer #2: Yes

4. Is the manuscript presented in an intelligible fashion and written in standard English?

Reviewer #1: Yes

Reviewer #2: Yes

5. Review Comments to the Author

Reviewer #1: Reviewer’s Comments

Thank you for the opportunity to review this manuscript which reports on the findings of a complex sample survey analysis conducted to determine the prevalence and factors associated with intention to accept COVID-19 vaccine among internally displaced persons (IDPs) living in Northwest Nigeria. Given the under-representation of IDPs in research studies with this focus and their general exclusion from COVID-19 mitigation programs at the height of the pandemic globally, the findings of this study should make an important contribution to informing equitable and responsive health care provision to this population. The manuscript is well written and presented. There are some aspects of the methodology and reporting which requires clarity.

Major Comments

To provide further context to the study, the authors should address – as part of the Background section - the burden of COVID-19 in Nigeria (e.g., cases, hospitalization and/or deaths) and provide clarity on the national COVID-19 vaccination program, including access and provision of COVID-19 vaccines among marginalized populations (e.g., IDPs) where possible.

With regards to the Methods, there is a need for further clarity on the types of data collected from the household surveys conducted in July – December 2023 and whether this survey is separate from the semi-structured interviews conducted in the current study. It is unclear from the methods whether the study design adopted was prospective or retrospective. In addition, it is unclear to me when the study activities were conducted (i.e., when were participants recruited and enrolled into this study? What was the duration of this study?).

While the comparisons made with findings from general populations from other countries are notable in the Discussion section, the authors are encouraged to also appraise their findings against similar studies conducted among migrant populations (whether IDPs or otherwise) in other parts of the world. For example, you would note from a recently published systematic review and meta-analysis that the findings on vaccine acceptance in this study are comparable, if not higher, than that reported previously among migrant populations globally [Hajissa K, Mutiat HA, Kaabi NA, et al. COVID-19 Vaccine Acceptance and Hesitancy among Migrants, Refugees, and Foreign Workers: A Systematic Review and Meta-Analysis. Vaccines (Basel). 2023;11(6):1070. Published 2023 Jun 6. doi:10.3390/vaccines11061070].

Minor Comments

• Line 38, page 2; “…were 18–39 years old, 82.6.0% were females…” Kindly confirm the proportion presented here.

• Line 88, page 5; “Furthermore, a recent, unpublished study conducted among IDPs…” Kindly note the journal guidelines on this and address appropriately: "Do not cite the following sources in the reference list: Unavailable and unpublished work, including manuscripts that have been submitted but not yet accepted (e.g., “unpublished work,” “data not shown”). Instead, include those data as supplementary material or deposit the data in a publicly available database."

• Line 126, page 6; “…and who had not received any dose of COVID-19 vaccine.” Kindly indicate how vaccination history was determined (participant recall or valid vaccination records?).

• Line 127 – 128, page 6; “Data of these participants were extracted from a complex sample, household survey…” For clarity, what type of participant data were extracted from the household survey? I'm struggling to reconcile the household survey conducted in July – December 2023, and the semi-structured interviews described as part of the Data collection and data instrument sub-section. The study is presented here as a prospective one with the retrospective approach only mentioned in the Limitations section. The authors should explicitly state the study design adopted as part of the methods section and avoid ambiguity.

• Lines 148 – 149, page 7; “…data for the household survey via face-to-face interviews with study participants…” Where were these interviews conducted. I ask this with relation to the privacy of the interviews conducted.

• Line 149, page 7; “…using semi-structured data collection instrument.” Kindly provide further clarity on the semi-structured data collection instrument used. If adapted from a publicly available source, please cite appropriately. If not, kindly provide a template as part of supplementary material for ease of reference.

• Line 182, page 10; “Nigerian Nigeria (13,300 NGN = 30 US Dollars)” This is unclear to me. Are the authors referring to the Nigerian currency?

• Line 237, page 13; “…we obtained verbal informed consent from the participants…” As part of the ethics statement, kindly provide further clarification as to why written informed consent was not possible or required for this study.

• Table 3, page 16; The number of female and male participants in Table 1 and 2 do not correlate. Kindly review all data presented in the manuscript and ensure that they are in fact accurate.

• Lines 371 – 372, page 23; “This interpretation suggests that by inducing a higher perception of COVID-19 severity…” With regards to the recommendations made here and later on in the manuscript, I’m wondering if the authors would consider re-framing this to advocate for context-specific and culturally appropriate education campaigns tailored to the needs of IDPs in order to enhance ownership and informed decision-making on COVID-19 vaccination rather than inducing a higher perception of COVID-19 severity.

• Line 375, page 24; “…communication to shape disease risk reception…” Do the authors mean, "...disease risk perception..."?

Reviewer #2: This study focused on determining the intention of internally displaced persons (IDPs) to accept COVID-19 vaccination and the factors associated with their choice as communicated by the IDPs involved in the study. The authors indicated that about 63.5% of IDPs expressed an intent to accept the COVID-19 vaccine while less than 2/3rd of the IDPs expressed no intent to accept the COVID-19 vaccine. Reasons behind the IDPs choice to accept the COVID-19 vaccine included a perception of COVID-19 as a severe disease and the living of IDPs in official or formal camps while the reasons behind the IDPs choice to reject the COVID-19 vaccine included COVID-19 misconceptions, vaccine safety concerns, did not feel the need or had no reasons at all.

The reviewer believes the overall information presented in the manuscript is important and should be accepted for publication after minor revisions indicating below.

1) The authors needs to re-word the introduction part of the manuscript from line 73 onwards to better reflect the current situation of COVID-19 globally and in Nigeria. The way they have written the introduction portion currently comes across like COVID-19 is still a global pandemic and the world is in lockdown. But COVID-19 pandemic is now properly managed especially in the developing countries.

Hence, the authors need to indicate the current COVID-19 situation globally and most importantly indicate the situation in developing countries especially in Nigeria where the study was conducted.

2) In the study setting, the authors gave a detailed description of the different camps situation aka formal versus informal camps. However, they did not indicate whether the IDPs included and interviewed for the study lived in official or unofficial camps or a mixture of both. The authors should therefore clearly indicate this and include the information in their results section and their study participants and data source section.

3) The authors also indicated that 63.5% of IDPs express an intent to accept COVID-19 vaccination but it is not clearly stated how this percentage was calculated or what numbers were used to generate this percentage as the numbers indicated in Table 3 does not equate to 63.5% of IDPs. Thus, the authors need to specify how they obtained the 63.5% indicated in the manuscript. Also, the authors need to indicate the numbers used to obtain this percentage. Furthermore, the authors need to indicate the percentage of IDPs that express an intent to reject the COVID-19 vaccine.

In Table 3, the percentage of each of the features indicated are calculated based on the numbers of IDPs that express and intent to either accept or reject the COVID-19 vaccine. Hence, in the results text description, the authors need to make this clearer as it is not obvious.

4) In the discussion section, paragraph 3 contains some redundant information (e.g., misconception, vaccine safety concerns, no felt need, etc.) that had already been stated in paragraph 1. The authors need to consolidate the information in the two paragraphs into one paragraph.

5) Again, the authors need to indicate whether the perceived COVID-19 severity and vaccine effectiveness were obtained from IDPs living in formal or informal camps

6) The authors do not clearly state that over 50% of IDPs express an intent to accept the COVID-19 vaccine but rather they try to belittle this data by giving ambigous percentage of the number of IDPs that expressed an intent to reject the vaccine. They do however, provide a good reference of the percentage of IDPs that expressed an intent to accept the COVID-19 vaccine being below the percentage seen in other generalized populations.

The authors are advised to provide clear and unbiased information regarding the percentage value of IDPs that express an intent to accept or reject the COVID-19 vaccine and not play down the fact that over 50% of IDPs did express an intent to take the COVID-19 vaccine.

7) When the authors describe the factors affiliated with the IDPs independent decission to take or reject the COVID-19 vaccine, they need to be very clear that the percentage quoted is based on the number of individuals tabulated in each category (e.g., intention to express the COVID-19 vaccine) and not the number of individuals in the total population.

8) Finally, the authors need to provide a bit more description about their statistically analyses performed for this study.

6. PLOS authors have the option to publish the peer review history of their article (what does this mean?). If published, this will include your full peer review and any attached files.

Reviewer #1: **Yes: **Edina Amponsah-Dacosta

Reviewer #2: No

---

## [Author Response · Author response to Decision Letter 0]

23 Apr 2024

Responses to Academic Editor’s Comments

Comment 

The manuscript is well-written and timely, particularly given the invaluable insights garnered from navigating the challenges posed by the COVID-19 pandemic. It is imperative to document such experiences, especially concerning vulnerable populations, to inform future healthcare strategies effectively.

Response

We sincerely appreciate the academic editor for these kind comments and inspiring remarks, and for highlighting the public health significance of our research. 

Comment 

However, to provide a comprehensive understanding of the paper's context, it is essential to delineate the impact of the COVID-19 pandemic in Africa and Nigeria, particularly concerning morbidity, mortality, vaccination rates, and other pertinent information. This comparative analysis would elucidate the specific challenges faced by internally displaced persons (IDPs) and set the backdrop for the study's relevance.

Response

We acknowledge the insight provided by the academic editor. Accordingly, we have revised the background of the study to reflect the editors’ useful submission and advice (Lines 54 – 68). 

Comment 

While the paper employs various statistical methodologies, it lacks clarity regarding the percentage of IDPs, regardless of their formal or informal status, willing to accept the COVID-19 vaccine. 

Response

The proportion of internally displaced persons (IDPs) willing to accept COVID-19 vaccine was 63.5% (95% CI: 59.0 – 68.1). We presented this proportion in the result section of the abstract (Lines 36 – 37) and the result section of the body of the manuscript ( Lines 268 – 269). This proportion is a “weighted proportion” obtained from the complex sample survey analysis, which was performed to account for the unequal probabilities of selecting participants via a complex sample design rather than a simple random sampling technique. We have elaborated on this pertinent point in the manuscript (Lines 200 – 207).

Comment 

Additionally, the issue of proximity to health facilities was highlighted, warranting further exploration, particularly in comparison to the general population. Also as COVID 19 is no longer a pandemic, can this also be reflected in the paper.

Response

As rightly indicated by the academic editor, the study revealed that IDPs who resided 2 kilometers or farther from the nearest health facility were less likely to accept COVID-19 vaccine. This important finding was extensively discussed in paragraph 7 of the discussion section (Lines 418 – 433). We elaborated on possible explanations for this finding in line with the data from our study. Moreover, we compared this result to the findings of several other studies that reported distance as a major barrier to the uptake of vaccination and other healthcare services at health facilities, even among the general population (Lines 425 – 427). Based on the evidence from our study, which resonates with findings from other similar studies, we highlighted the need for appropriate interventions, including the use of mobile vaccination strategy, to reach and deliver vaccines to IDPs in remote households (Lines 431 – 433). Furthermore, we recognize that COVID-19 is no longer considered a public health emergency; we have reflected this in the manuscript.

Comment 

Further elaboration on the statistical methodologies employed would enhance the paper's accessibility, especially for readers with limited statistical proficiency. Clarification and additional information on these methodologies would mitigate confusion and facilitate a deeper understanding of the findings.

Response

We acknowledge the concern of the academic editor. The comprehensive statistical approaches employed in this study were aimed at enhancing the validity of the study results, particularly because of the complex sample design approach. However, in accordance with the academic editor’s comment, we have provided additional information to enhance the clarity of the statistical methodologies we adopted in the study. Additionally, we have indicated appropriate references for the statistical approaches where necessary (Lines 200 – 221). 

Comment 

Lastly, it is advisable to limit references to publicly available "published work" and consider the feedback provided by Dr. Edina and Bassey. Their insights could further refine and strengthen the manuscript.

Response

We appreciate this useful advice. We have limited all our references to only “published work.”

Responses to Reviewers’ Comments

Reviewer 1

Comment 

Thank you for the opportunity to review this manuscript which reports on the findings of a complex sample survey analysis conducted to determine the prevalence and factors associated with intention to accept COVID-19 vaccine among internally displaced persons (IDPs) living in Northwest Nigeria. Given the under-representation of IDPs in research studies with this focus and their general exclusion from COVID-19 mitigation programs at the height of the pandemic globally, the findings of this study should make an important contribution to informing equitable and responsive health care provision to this population. The manuscript is well written and presented. 

Response

We are sincerely grateful to the reviewer for the important remarks and the meticulous review of our paper. 

Major Comments

Comment 

To provide further context to the study, the authors should address – as part of the Background section - the burden of COVID-19 in Nigeria (e.g., cases, hospitalization and/or deaths) and provide clarity on the national COVID-19 vaccination program, including access and provision of COVID-19 vaccines among marginalized populations (e.g., IDPs) where possible.

Response

We thank the reviewer for this comment and the useful suggestion. Accordingly, we have included this relevant information in the introduction section of the revised manuscript (Lines 54 – 68).

Comment 

With regards to the Methods, there is a need for further clarity on the types of data collected from the household surveys conducted in July – December 2023 and whether this survey is separate from the semi-structured interviews conducted in the current study. It is unclear from the methods whether the study design adopted was prospective or retrospective. In addition, it is unclear to me when the study activities were conducted (i.e., when were participants recruited and enrolled into this study? What was the duration of this study?).

Response

These are pertinent comments. To provide additional clarity on the study design, study participants, and the types of data, we have substantially revised the relevant sub-sections of the “Materials and Methods” section of the manuscript (Lines 130 – 163). Furthermore, the description and categories of study variables are presented in Table 1. 

Comment 

While the comparisons made with findings from general populations from other countries are notable in the Discussion section, the authors are encouraged to also appraise their findings against similar studies conducted among migrant populations (whether IDPs or otherwise) in other parts of the world. For example, you would note from a recently published systematic review and meta-analysis that the findings on vaccine acceptance in this study are comparable, if not higher, than that reported previously among migrant populations globally [Hajissa K, Mutiat HA, Kaabi NA, et al. COVID-19 Vaccine Acceptance and Hesitancy among Migrants, Refugees, and Foreign Workers: A Systematic Review and Meta-Analysis. Vaccines (Basel). 2023;11(6):1070. Published 2023 Jun 6. doi:10.3390/vaccines11061070].

Response

We sincerely acknowledge this useful comment. Although there is a dearth of similar studies conducted among internally displaced persons, refugees, and other migrant populations, we have expanded the scope of our discussion to compare the findings of our study with the results of the study mentioned by the reviewer and with the few similar research conducted among refugees and migrant populations in the published literature (Lines 344 – 349). 

Minor Comments

Comment 

Line 38, page 2; “…were 18–39 years old, 82.6.0% were females…” Kindly confirm the proportion presented here.

Response

We thank the reviewer for this observation. The proportion was 82.6%; we have corrected this in the manuscript (Line 36).

Comment 

Line 88, page 5; “Furthermore, a recent, unpublished study conducted among IDPs…” Kindly note the journal guidelines on this and address appropriately: "Do not cite the following sources in the reference list: Unavailable and unpublished work, including manuscripts that have been submitted but not yet accepted (e.g., “unpublished work,” “data not shown”). Instead, include those data as supplementary material or deposit the data in a publicly available database."

Response

We noted this comment with gratitude. We have included the citation for the referenced source and reworded the sentence (Lines 98 – 101).

Comment 

Line 126, page 6; “…and who had not received any dose of COVID-19 vaccine.” Kindly indicate how vaccination history was determined (participant recall or valid vaccination records?).

Response

The vaccination status was determined via participant’s recall (or history). We have indicated this in the manuscript (Lines 134 – 136).

Comment 

Line 127 – 128, page 6; “Data of these participants were extracted from a complex sample, household survey…” For clarity, what type of participant data were extracted from the household survey? I'm struggling to reconcile the household survey conducted in July – December 2023, and the semi-structured interviews described as part of the Data collection and data instrument sub-section. The study is presented here as a prospective one with the retrospective approach only mentioned in the Limitations section. The authors should explicitly state the study design adopted as part of the methods section and avoid ambiguity.

Response

We sincerely thank the reviewer for this useful feedback. In line with this suggestion, we have revised the “Materials and Methods” section of the manuscript to provide clarity on the study design, study participants, and data sources ({Lines 130 – 163} and Table 1).

Comment 

Lines 148 – 149, page 7; “…data for the household survey via face-to-face interviews with study participants…” Where were these interviews conducted. I ask this with relation to the privacy of the interviews conducted.

Response

This question is apt. The interviews were conducted within the households to maintain privacy and confidentiality. We have revised the sentence to reflect this information (Lines 147 – 148).

Comment 

Line 149, page 7; “…using semi-structured data collection instrument.” Kindly provide further clarity on the semi-structured data collection instrument used. If adapted from a publicly available source, please cite appropriately. If not, kindly provide a template as part of supplementary material for ease of reference.

Response

We have revised this section of the manuscript. The current study extracted data from a household complex sample survey, which was described extensively in a published article. We have cited this publication in the appropriate section of the revised paper (Lines 131 – 134) and briefly described the sampling methodology and data collection procedure of the household complex sample survey (Lines 140 – 149).

Comment 

Line 182, page 10; “Nigerian Nigeria (13,300 NGN = 30 US Dollars)” This is unclear to me. Are the authors referring to the Nigerian currency?

Response

Yes, NGN is the currency code for Nigerian Naira. We also indicated this in the footnotes of the respective Tables (Lines 171, 259, 278, and 327 ).

Comment 

Line 237, page 13; “…we obtained verbal informed consent from the participants…” As part of the ethics statement, kindly provide further clarification as to why written informed consent was not possible or required for this study.

Response

We acknowledge this important comment. In accordance with the reviewer’s advice, we have provided additional information regarding the informed consent (Lines 233 – 236).

Comment 

Table 3, page 16; The number of female and male participants in Table 1 and 2 do not correlate. Kindly review all data presented in the manuscript and ensure that they are in fact accurate.

Response

We thank the reviewer for bringing this error to our attention. We have made the necessary corrections in Table 3 (Page 16) and double-checked all the data and results in the manuscript for accuracy. 

Comment 

Lines 371 – 372, page 23; “This interpretation suggests that by inducing a higher perception of COVID-19 severity…” With regards to the recommendations made here and later on in the manuscript, I’m wondering if the authors would consider re-framing this to advocate for context-specific and culturally appropriate education campaigns tailored to the needs of IDPs in order to enhance ownership and informed decision-making on COVID-19 vaccination rather than inducing a higher perception of COVID-19 severity.

Response

We are grateful for this invaluable suggestion. Accordingly, we have revised the appropriate sections of the manuscript to reflect the reviewer’s advice (Lines 382 – 387 and Lines 455 – 460).

Comment 

Line 375, page 24; “…communication to shape disease risk reception…” Do the authors mean, "...disease risk perception..."?

Response

Yes, we meant to write “disease risk perception.” We have corrected this in the paper (Line 385).

Reviewer 2

Comment 

1) The authors needs to re-word the introduction part of the manuscript from line 73 onwards to better reflect the current situation of COVID-19 globally and in Nigeria. The way they have written the introduction portion currently comes across like COVID-19 is still a global pandemic and the world is in lockdown. But COVID-19 pandemic is now properly managed especially in the developing countries. Hence, the authors need to indicate the current COVID-19 situation globally and most importantly indicate the situation in developing countries especially in Nigeria where the study was conducted.

Response

We thank the reviewer for these important comments. In line with the reviewer’s advice, we have revised the introduction section of the manuscript to reflect the current COVID-19 situation globally, in Africa, and in Nigeria (Lines 54 – 68).

Comment 

2) In the study setting, the authors gave a detailed description of the different camps situation aka formal versus informal camps. However, they did not indicate whether the IDPs included and interviewed for the study lived in official or unofficial camps or a mixture of both. The authors should therefore clearly indicate this and include the information in their results section and their study participants and data source section.

Response

The study participants included in this study resided in both formal and informal IDPs camps across Borno, Adamawa, and Yobe States (BAY States). We have included this information in the study design and data source sub-section of the manuscript (Lines 134 – 136). Furthermore, the results of COVID-19 vaccination intention (Table 3) and factors associated with intention to accept COVID-19 (Table 5) included data on the status of IDPs camps (formal versus informal).

Comment 

3) The authors also indicated that 63.5% of IDPs express an intent to accept COVID-19 vaccination but it is not clearly stated how this percentage was calculated or what numbers were used to generate this percentage as the numbers indicated in Table 3 does not equate to 63.5% of IDPs. Thus, the authors need to specify how they obtained the 63.5% indicated in the manuscript. Also, the authors need to indicate the numbers used to obtain this percentage. Furthermore, the authors need to indicate the percentage of IDPs that express an intent to reject the COVID-19 vaccine.

In Table 3, the percentage of each of the features indicated are calculated based on the numbers of IDPs that express and intent to either accept or reject the COVID-19 vaccine. Hence, in the results text description, the authors need to make this clearer as it is not obvious.

Response

We acknowledge this important observation. Data of the participants included in th

---

## [Decision Letter · Decision Letter 1]

16 May 2024

PONE-D-24-02466R1COVID-19 vaccination intention among internally displaced persons in complex humanitarian emergency context, Northeast NigeriaPLOS ONE

Dear Dr. Gidado,

Thank you for submitting your manuscript to PLOS ONE. After careful consideration, we feel that it has merit but does not fully meet PLOS ONE’s publication criteria as it currently stands. Therefore, we invite you to submit a revised version of the manuscript that addresses the points raised during the review process.

We look forward to receiving your revised manuscript.

Kind regards,

Sylvester Maleghemi, MD,MSC, DTH&M, MBA

Academic Editor

PLOS ONE

Journal Requirements:

Additional Editor Comment:

Dear Authors'

Thank you for your thorough revisions and for addressing most of the concerns the reviewers raised. However, there are a few remaining points from Reviewer 2's comments that require your attention for minor revisions:

Overall Quality:

• The manuscript is well-written, timely, and offers valuable insights into the vaccination intentions of a vulnerable population, which is crucial for informing public health strategies. The statistical analysis is robust and appropriate for the study design.

Major Comments:

1. Clarification of Study Design and Methods:

• The study design and data collection methods need further clarification. Specifically, there should be a clear distinction between the household surveys conducted and the semi-structured interviews. It is also important to specify whether the study was prospective or retrospective and to include precise dates for participant recruitment and study duration.

2. Calculation and Presentation of Statistics:

• The authors should provide detailed information on how the 63.5% intention to accept COVID-19 vaccination was calculated. This includes specifying the numbers and methodology used to derive this percentage, as the numbers in Table 3 do not straightforwardly equate to this figure. Additionally, ensure that the percentage of IDPs who intend to reject the vaccine is also clearly stated in the manuscript.

3. Contextualization of COVID-19 Situation:

• The introduction should be reworded to better reflect the current global and Nigerian context of COVID-19. This should include the current status of COVID-19 as no longer a pandemic and the specific situation in Nigeria, emphasizing how the management of COVID-19 has evolved.

4. Comparison with Other Populations:

• The discussion should include comparisons with similar studies conducted among migrant populations, such as refugees and other displaced groups globally. For instance, comparing the findings with those from a recent systematic review and meta-analysis on vaccine acceptance among migrant populations would provide a broader context and relevance.

Minor Comments:

1. Typographical and Formatting Corrections:

• Line 182: Correct "Nigerian Nigeria" to "Nigerian Naira".

• Line 375: Correct "disease risk reception" to "disease risk perception".

2. Additional Details in Methods:

• Provide more details about the semi-structured data collection instrument, including whether it was adapted from an existing source or developed for this study. If adapted, cite the source; if developed, include a template in the supplementary material.

3. Privacy During Data Collection:

• Indicate where the interviews were conducted, ensuring the privacy and confidentiality of the participants. This is particularly important given the sensitive nature of the information collected.

Specific Revisions Required:

1. Include Weighted Percentage of IDPs Rejecting Vaccine:

• As noted by the reviewer, state the weighted percentage of IDPs intending to reject the COVID-19 vaccine in the section encompassing lines 280-283, similar to how the intention to accept the vaccine is stated.

2. Include Response Text in Methods and Results:

• Incorporate parts of the text from the rebuttal/response to comment 3 of reviewer 2 into the methods section under the data analysis subheading. This should explain the rationale behind the data collection approach and include details on the statistical analysis method. Additionally, include text from the response to comment 5 of reviewer 2 in the appropriate results section.

3. Correct Currency Terminology:

• Correct the terminology in Line 171 to "Nigerian Naira" and ensure this is consistently used throughout the manuscript.

Conclusion:

Overall, the manuscript provides significant contributions to understanding COVID-19 vaccination intentions among IDPs in Northeast Nigeria. By addressing the above comments, the authors can enhance the clarity, accuracy, and impact of their study. These revisions will help ensure the manuscript meets the high standards expected for publication.

Reviewers' comments:

Reviewer's Responses to Questions

**Comments to the Author**

1. If the authors have adequately addressed your comments raised in a previous round of review and you feel that this manuscript is now acceptable for publication, you may indicate that here to bypass the “Comments to the Author” section, enter your conflict of interest statement in the “Confidential to Editor” section, and submit your "Accept" recommendation.

Reviewer #1: All comments have been addressed

Reviewer #2: (No Response)

2. Is the manuscript technically sound, and do the data support the conclusions?

Reviewer #1: Yes

Reviewer #2: Yes

3. Has the statistical analysis been performed appropriately and rigorously? 

Reviewer #1: Yes

Reviewer #2: Yes

4. Have the authors made all data underlying the findings in their manuscript fully available?

Reviewer #1: Yes

Reviewer #2: Yes

5. Is the manuscript presented in an intelligible fashion and written in standard English?

Reviewer #1: Yes

Reviewer #2: Yes

6. Review Comments to the Author

Reviewer #1: Thank you for taking the time to carefully consider the comments and suggestions raised during the first round of review. I have the following final issues which need addressing:

Comment: Line 182, page 10; “Nigerian Nigeria (13,300 NGN = 30 US Dollars)” This is unclear to

me. Are the authors referring to the Nigerian currency?

Response: Yes, NGN is the currency code for Nigerian Naira. We also indicated this in the

footnotes of the respective Tables (Lines 171, 259, 278, and 327 ).

This has still not been revised to Nigerian Naira as raised during the previous review. Please revise this throughout the manuscript where it appears.

Lines 394 - 395, page 24; "Although the safety and effectiveness of COVID-19 vaccine are widely disputed..." To avoid this statement being misconstrued, I would encourage the authors to qualify what they mean by "widely disputed". Does this imply that the scientific expert community is uncertain about the safety and effectiveness of all COVID-19 vaccines or does this refer to the uncertainties prominent among the general public?

Reviewer #2: The authors have greatly revised the manuscript and addressed a large proportion of the raised concerns after reviewing the original manuscript. However, some concerns were not fully addressed and require minor revisions.

The authors did not respond fully to comment (3) of reviewer 2 and should therefore state the weighted % of IDPs that expressed an intent to reject the COVID-19 vaccines in the section encompassing lines 280 - 283 just as they stated for the IDPs who expressed a desire to accept COVID-19 vaccine as indicated in lines 268 - 269 of their revised manuscript. The numbers are already presented in their table so I don't assume it would be difficult to use their statistical analysis method to obtain the requested result.

Also, the authors need to include some of the text in their rebuttal/response to comment (3) of reviewer 2 in the methods section of the revised manuscript text - specifically the data analysis subheading section. The authors should start by explaining simply why they decided to collect the data for this study using a complex approach as it will provide some context to some of the method description in the data analysis section of their revised manuscript before including where appropriate some of their response to comment 3 as indicated above in the revised manuscript methods data analysis section.

Also some of the response text to comment (5) of reviewer 2 should be include in the appropriate results sectionof the revised manuscript text.

Finally, the authors need to correct the error in Line 171 Nigerian Nigeria to Nigerian Naira as indicated in lines 259, 278 and 327.

7. PLOS authors have the option to publish the peer review history of their article (what does this mean?). If published, this will include your full peer review and any attached files.

Reviewer #1: **Yes: **Edina Amponsah-Dacosta

Reviewer #2: No

---

## [Author Response · Author response to Decision Letter 1]

2 Jul 2024

Responses to Additional Comments by the Journal and the Reviewers 

Journal Requirements:

Comment 

Please review your reference list to ensure that it is complete and correct. If you have cited papers that have been retracted, please include the rationale for doing so in the manuscript text or remove these references and replace them with relevant current references. Any changes to the reference list should be mentioned in the rebuttal letter that accompanies your revised manuscript. If you need to cite a retracted article, indicate the article’s retracted status in the References list and also include a citation and full reference for the retraction notice.

Response

We are grateful for this comment. We have reviewed the reference list for completeness and appropriateness. Accordingly, we have replaced references 2, 6, 10, and 31 in the reference list of the revised manuscript. 

Additional Editor Comments:

Dear Authors'

Thank you for your thorough revisions and for addressing most of the concerns the reviewers raised. However, there are a few remaining points from Reviewer 2's comments that require your attention for minor revisions.

Response

We are grateful for these kind words and happy to address all outstanding reviewers’ comments 

Overall Quality

Comment

The manuscript is well-written, timely, and offers valuable insights into the vaccination intentions of a vulnerable population, which is crucial for informing public health strategies. The statistical analysis is robust and appropriate for the study design.

Response

We thank the reviewer for these encouraging remarks. 

Major Comments

Comment

1. Clarification of Study Design and Methods:

• The study design and data collection methods need further clarification. Specifically, there should be a clear distinction between the household surveys conducted and the semi-structured interviews. It is also important to specify whether the study was prospective or retrospective and to include precise dates for participant recruitment and study duration.

Response

We appreciate the reviewer for these comments. In line with the reviewer’s suggestion, we have revised the “study design and data source” section of the manuscript to enhance clarity. The current study was a cross-sectional study examining exposure and outcome variables at the same point in time. The precise date of data collection has been included in the revised manuscript. 

 In the revised manuscript, this section reads in part: 

“This cross-sectional study included a total of 1,537 IDPs aged 18 years and above, spread across 18 formal and informal IDPs camps in Borno, Adamawa, and Yobe States (BAY States) in Nigeria’s northeast region. These IDPs were among the participants enrolled from July 25 to December 5, 2022, in a complex sample household survey to investigate COVID-19 knowledge, risk perception, and adherence to COVID-19 preventive measures among IDPs in the BAY States, Northeast Nigeria. Due to practical and logistic considerations, participants enrolled in the complex sample household survey were sampled using a complex sampling methodology. As of their enrolment in the complex sample household survey, the IDPs included in the current study had not received any dose of COVID-19 vaccine based on history (self-report). The socio-demographic and household data of these 1,537 IDPs, as well as data regarding their COVID-19 knowledge, risk perception, and vaccination intention, were extracted from the database of the complex sample household survey. The sampling methodology and data collection approach for the complex sample household survey have been described elsewhere…… Data for the complex sample household survey were collected using a semi-structured data collection instrument uploaded on Open Data Kit (ODK). This data instrument is included as Supporting Information S1 File.” 

Comment

2. Calculation and Presentation of Statistics:

• The authors should provide detailed information on how the 63.5% intention to accept COVID-19 vaccination was calculated. This includes specifying the numbers and methodology used to derive this percentage, as the numbers in Table 3 do not straightforwardly equate to this figure. Additionally, ensure that the percentage of IDPs who intend to reject the vaccine is also clearly stated in the manuscript.

Response

We appreciate the reviewer for this comment. The 63.5% intention to accept COVID-19 vaccine was computed using a complex sample data analysis methodology. As we detailed in the first round of reviews, the socio-demographic and household data of the 1,537 participants included in the current study, as well as data regarding their COVID-19 knowledge, risk perception, and vaccination intention, were extracted from the database of a complex sample household survey. Because the participants enrolled in the complex sample household survey were recruited via a complex sampling approach rather than a simple random sampling technique, we performed a complex sample survey analysis to account for the differential (unequal) probabilities of selecting these participants in line with the recommended data analysis approach for complex sampling surveys. Typically, complex sample data analysis yields weighted statistical estimates, proportions, standard errors, and confidence intervals. 

In the current study, the absolute number of participants who expressed intention to accept COVID-19 vaccine was 1,105 (Line 271). Ordinarily, the non-weighted proportion expected from a non-complex sample design analysis is 1,105/1,537 = 71.9%. However, for the purpose of complex sample analysis, the response of each of the 1,537 participants was adjusted for the survey weight each participant contributed to the analysis based on the sampling probabilities of these participants. The survey weight was obtained by determining the inverse of the sampling probabilities. Each participant's response was thereafter multiplied by the survey weight. For instance, a participant with a higher sampling probability has a lower survey weight than a participant with a lower sampling probability, and vice-versa. In simple terms, each participant’s response was multiplied by the survey weight to obtain a weighted response. The weighted responses of all participants were then summed up and divided by the total population of IDPs to obtain a weighted proportion. This computation was seamlessly executed using the “survey package” of R statistical and computing software, which we utilized for data analysis. 

In accordance with the standard statistical procedure for complex sample analysis, we applied the principle of weighted analysis to the weighted logistic regression and other statistical analyses in this study. For the same reason, we used the Rao-Scott chi-square test, a design-adjusted version of the Pearson chi-square test, to determine the relationship between vaccination intention and key participants’ characteristics instead of the conventional Pearson chi-square. In the revised manuscript, we provided appropriate references for publications that describe the conduct of complex sample data analysis.

All the proportions we presented in Table 3 were weighted – the same for all other results we presented in the result tables and elsewhere in the manuscript. 

The weighted proportion of participants who intend to reject vaccines was 36.5% (95% CI: 31.9 – 41.0). In line with the reviewer's suggestion, we have included this weighted proportion and the 95% confidence interval in the manuscript. This section now reads as follows:

“Of the 1,537 participants, 1,105 expressed intention to accept COVID-19 vaccine, corresponding to a weighted proportion of 63.5% (95% CI: 59.0 – 68.1), whereas 432 participants, with a weighted proportion of 36.5% (95% CI: 31.9 – 41.0) reported intention to reject COVID-19 vaccine.” 

 Comment

3. Contextualization of COVID-19 Situation:

• The introduction should be reworded to better reflect the current global and Nigerian context of COVID-19. This should include the current status of COVID-19 as no longer a pandemic and the specific situation in Nigeria, emphasizing how the management of COVID-19 has evolved.

Response

We appreciate the reviewer for this comment. In response to a similar comment by the academic editor and Reviewer 1 in the first round of review, we have substantially edited the introduction section of the manuscript to provide appropriate context for our study (Lines 54 – 115). With relevant citations, we provided an elaborate background of the identification, emergence, and situation of COVID-19 globally and in Nigeria. We indicated that the World Health Organization declared an end to COVID-19 global health emergency on May 5, 2023. We described the COVID-19 risk profiles for IDPs and justified the need to optimize COVID-19 vaccination uptake among this population. Citing relevant scientific publications, we noted that although COVID-19 is no longer considered a public health emergency, there is a risk of the emergence of new variants that could be more transmissible and/or more severe with new surges in cases and deaths. Amidst the controversy that trailed the safety and effectiveness of COVID-19 vaccine worldwide, we provided relevant background information regarding the acceptance of COVID-19 vaccine globally, in Nigeria, and among the IDPs in northeast Nigeria. Furthermore, we highlighted the dearth of research on COVID-19 vaccination intention among IDPs in humanitarian emergency contexts to further justify the conduct of our study. We cited publications that asserted that findings of research conducted among different population sub-groups in stable, non-humanitarian contexts cannot simply be extrapolated to inform disease prevention and control interventions in humanitarian emergency situations. We believe the edited introduction of the revised manuscript appropriately contextualized our study, provided essential background information, and highlighted the rationale for the conduct of the study. 

Comment

4. Comparison with Other Populations:

• The discussion should include comparisons with similar studies conducted among migrant populations, such as refugees and other displaced groups globally. For instance, comparing the findings with those from a recent systematic review and meta-analysis on vaccine acceptance among migrant populations would provide a broader context and relevance.

Response

This is a pertinent comment. In line with the reviewer’s suggestion, we have expanded the scope of our discussion to compare our findings with the results of the study mentioned by the reviewer and the results of other similar research conducted among internally displaced persons, refugees, and other migrant populations published in the scientific literature. 

Minor Comments:

Comment

1. Typographical and Formatting Corrections:

• Line 182: Correct "Nigerian Nigeria" to "Nigerian Naira".

Response

We appreciate the reviewer for pointing out this error. We have corrected the error in the revised manuscript.

Comment

• Line 375: Correct "disease risk reception" to "disease risk perception".

Response

We have corrected this in the revised manuscript (Line 395) 

Comment

2. Additional Details in Methods:

• Provide more details about the semi-structured data collection instrument, including whether it was adapted from an existing source or developed for this study. If adapted, cite the source; if developed, include a template in the supplementary material.

Response

The semi-structured data collection instrument was developed by the research team to collect data for the complex sample household survey described in the manuscript (Lines 144 – 154). The items in the instrument were informed by a review of the literature on similar subjects. Furthermore, the items were appropriately contextualized to reflect the educational level of the study participants and the peculiarities of the research setting. Since the 1,537 IDPs included in the current study were among the participants enrolled in the complex sample household survey, we have included a template of the data collection instrument for the complex sample household survey in the supplementary materials, in line with the suggestion of the reviewer. 

Comment

3. Privacy During Data Collection:

• Indicate where the interviews were conducted, ensuring the privacy and confidentiality of the participants. This is particularly important given the sensitive nature of the information collected.

Response

All the participants enrolled in the complex sample household survey, including the 1,537 IDPs in the current study, were interviewed within the households to maintain privacy and confidentiality. We have included this information in the revised manuscript.

Specific Revisions Required:

Comment

1. Include Weighted Percentage of IDPs Rejecting Vaccine:

• As noted by the reviewer, state the weighted percentage of IDPs intending to reject the COVID-19 vaccine in the section encompassing lines 280-283, similar to how the intention to accept the vaccine is stated.

Response

We appreciate this suggestion. As advised, we have included the weighted percentage of IDPs intending to reject COVID-19 in the result section of the manuscript. This section now reads as follows:

“Of the 1,537 participants, 1,105 expressed intention to accept COVID-19 vaccine, corresponding to a weighted proportion of 63.5% (95% CI: 59.0 – 68.1), whereas 432 participants, with a weighted proportion of 36.5% (95% CI: 31.9 – 41.0) reported intention to reject COVID-19 vaccine.” 

Comment

2. Include Response Text in Methods and Results:

• Incorporate parts of the text from the rebuttal/response to comment 3 of reviewer 2 into the methods section under the data analysis subheading. This should explain the rationale behind the data collection approach and include details on the statistical analysis method. Additionally, include text from the response to comment 5 of reviewer 2 in the appropriate results section.

Response

We acknowledge this comment. In line with this suggestion, we have included parts of our initial response to comment 3 of Reviewer 2 and edited the “study design and data sources” section of the revised manuscript to enhance clarity. This section now reads: 

“This cross-sectional study included a total of 1,537 IDPs aged 18 years and above, spread across 18 formal and informal IDPs camps in Borno, Adamawa, and Yobe States (BAY States) in Nigeria’s northeast region. These IDPs were among the participants enrolled from July 25 to December 5, 2022, in a complex sample household survey to investigate COVID-19 knowledge, risk perception, and adherence to COVID-19 preventive measures among IDPs in the BAY States, Northeast Nigeria. Due to practical and logistic considerations, participants enrolled in the complex sample household survey were sampled using a complex sampling methodology. As of their enrolment in the complex sample household survey, the IDPs included in the current study had not received any dose of COVID-19 vaccine based on history (self-report). The socio-demographic and household data of these 1,537 IDPs, as well as data regarding their COVID-19 knowledge, risk perception, and vaccination intention, were extracted from the database of the complex sample household survey.” 

Furthermore, the details of our statistical analysis approach are provided under the data analysis and statistical methods section of the revised manuscript (Lines 205 – 227). As suggested, we have updated the result section of the manuscript with some of our earlier responses to comment 5 of Reviewer 2. The revised section of the results now reads:

“Adjusting for co-variates at multivariable logistic regression analysis, IDPs who perceived COVID-19 as a severe illness compared to those who did not (Adjusted Odds Ratio (AOR) = 2.31, [95% CI: 1.35 – 3.96]), and IDPs who perceived COVID-19 vaccine as effective compared to those who did not (AOR = 4.28, [95% CI: 2.46 – 7.44]), irrespective of whether they resided in formal or informal camps, were significantly more likely to accept COVID-19 vaccine.” 

Comment

3. Correct Currency Terminology:

• Correct the terminology in Line 171 to "Nigerian Naira" and ensure th

---

## [Decision Letter · Decision Letter 2]

15 Jul 2024

COVID-19 vaccination intention among internally displaced persons in complex humanitarian emergency context, Northeast Nigeria

PONE-D-24-02466R2

Dear Dr. Gidado,

We’re pleased to inform you that your manuscript has been judged scientifically suitable for publication and will be formally accepted for publication once it meets all outstanding technical requirements.

Kind regards,

Sylvester Maleghemi, MD, MSC, DTH&M, MBA

Academic Editor

PLOS ONE

Additional Editor Comments (optional):

The manuscript is well-written and clearly articulates the study's background, objectives, methodology, results, and implications. The statistical analysis is robust, and the findings are effectively contextualized within the existing literature. The revisions made in response to previous reviews have significantly improved the clarity and comprehensiveness of the manuscript.

The study design and data collection methods are well-detailed and appropriate for the research questions. The distinction between household surveys and semi-structured interviews is clear, and the study period is well-defined. Explaining the weighted percentage for IDPs intending to accept or reject the COVID-19 vaccine adds transparency to the methodology.

The statistical analysis is sound, using weighted logistic regression models appropriately. The interpretation of adjusted odds ratios is clear and well-supported by the data. Including the rationale for the statistical methods directly in the manuscript enhances transparency and reader understanding.

The authors have successfully addressed previous reviewer comments, particularly regarding comparisons with other populations and clarifications of the study design.

Thank you for the opportunity to review this work.

Reviewers' comments:

Reviewer's Responses to Questions

**Comments to the Author**

1. If the authors have adequately addressed your comments raised in a previous round of review and you feel that this manuscript is now acceptable for publication, you may indicate that here to bypass the “Comments to the Author” section, enter your conflict of interest statement in the “Confidential to Editor” section, and submit your "Accept" recommendation.

Reviewer #1: All comments have been addressed

2. Is the manuscript technically sound, and do the data support the conclusions?

Reviewer #1: Yes

3. Has the statistical analysis been performed appropriately and rigorously? 

Reviewer #1: I Don't Know

4. Have the authors made all data underlying the findings in their manuscript fully available?

Reviewer #1: Yes

5. Is the manuscript presented in an intelligible fashion and written in standard English?

Reviewer #1: Yes

6. Review Comments to the Author

Reviewer #1: The authors have satisfactorily addressed the queries I raised during the second round of reviews. I have no further comments.

7. PLOS authors have the option to publish the peer review history of their article (what does this mean?). If published, this will include your full peer review and any attached files.

Reviewer #1: **Yes: **Edina Amponsah-Dacosta

---

## [Editor Report · Acceptance letter]

23 Jul 2024

PONE-D-24-02466R2 

PLOS ONE

Dear Dr. Gidado, 

I'm pleased to inform you that your manuscript has been deemed suitable for publication in PLOS ONE. Congratulations! Your manuscript is now being handed over to our production team.

Kind regards, 

on behalf of

Dr. Sylvester Maleghemi Maleghemi 

Academic Editor

PLOS ONE